# The subjective experience of time during the pandemic in Germany: The big slowdown

**Ferdinand Kosak**[1]*, **Iris Schelhorn**[1], **Marc Wittmann**[2]

**1** Department of Psychology, University of Regensburg, Regensburg, Germany, **2** Institute for Frontier Areas of Psychology and Mental Health, Freiburg, Germany

* ferdinand.kosak@ur.de

## Abstract

Several COVID-19 studies on the felt passage of time have been conducted due to the strong feeling of time distortion many people have experienced during the pandemic. Overall, a relative deceleration of time passage was generally associated with negative affect and social isolation; a relative acceleration was associated with an increase in routine in daily life. There is some variability in results depending on the country of study and COVID-19 restrictions introduced, participants' demographics, and questionnaire items applied. Here we present a study conducted in May 2021 in Germany including n = 500 participants to assess time perception, emotional reactions, and attitudes towards the countermeasures. The passage of time judgments (POTJ) for the preceding 12 months during the pandemic were compared to data addressing the same question posed in previous studies conducted before the outbreak of COVID-19. The previous year was rated as having passed relatively slower during the pandemic compared to the ratings from before the pandemic. The duration judgments (DJ) of the 14 months since the start of the pandemic showed a bimodal distribution with both relatively shorter and relatively longer DJs. Higher levels of several negative emotions, as well as less social satisfaction, were associated with prolonged DJs and partially slower POTJs. Fear for health was not linked with the subjective experience of time, but exploratory analyses suggested that higher levels of fear were linked to more positive evaluations and approval of the governmental countermeasures. Those who reported higher levels of negative, agitated-aggressive emotions showed lower levels of consent with these measures.

**Data Availability Statement:** The data and analyses can be accessed via an online repository on OSF (https://osf.io/kc86r/).

**Funding:** The author(s) received no specific funding for this work.

## Introduction

Over the first weeks of social isolation during the COVID-19 pandemic in spring 2020, people typically reported that time had passed differently in this new situation—faster for some, but more slowly for others. This remarkable change in subjective time after a few weeks of social isolation was reported as significant time distortions in many newspapers around the world [1]. Time-perception researchers worldwide responded quickly and conducted online studies to assess potential changes in the experience of time during this crisis.

**Competing interests:** The authors have declared that no competing interests exist.

In an Italian study, 1310 individuals aged between 18 and 35 were asked to compare their experience during the first week of the lockdown with the week before the lockdown, when they still had a regular life [2]. Participants reported pronounced problems with keeping track of time, i.e., they were often confused about what time of the day or what day of the week it was. They also reported an expansion of time and this feeling was positively related with the sense of boredom. In two subsequent studies conducted in the UK, participants were asked how quickly the previous day and week had passed as compared to the time before the lockdown [3, 4]. The first study with 603 participants (on average 35 years old) assessed the experience between 14 and 38 days after the beginning of the lockdown. About 80% of participants reported unusual changes in their experience of time with a split into two subgroups: one reported experiences of the previous day/week as having passed slower and one as it having passed faster compared to before the pandemic. The second UK study [4] assessed experiences during the second lockdown 8 months later. 80% of the participants again reported a change in the experienced duration for the previous week or day compared to normal with a majority of participants rating the eight months since the beginning of the pandemic as having passed slower than usual.

In both UK studies, two main factors were identified which determined the felt passage of time: people who had increased stress levels and less satisfaction with their social situations judged time as having felt longer, while people who had lower stress levels and were more satisfied with their social situations reported intervals as shorter than before the pandemic.

Two French studies collected data during the first lockdown in March and April 2020 by assessing passage of time judgments, i.e., "What are your feelings about the speed of the passage of time?" [5, 6]. French participants characterized the time during the lockdown as an extreme increase in boredom and sadness; both of these feelings were related to the impression that time had passed more slowly during the lockdown compared to before the lockdown. The variable 'lack of activity' appeared to reliably explain the feeling of time slowing down. Follow-ups of these two studies 6 and 12 months later showed that this impression of a deceleration of time passage and its association with feelings of boredom and depression persisted [7].

In a longitudinal Brazilian study [8] with 3855 participants, the first assessment was undertaken 60 days after social-isolation measures had beein introduced. Weekly assessments of time awareness (measured with items, such as "I often think that time just does not want to pass.") followed for 14 weeks during the pandemic. This longitudinal design enabled researchers to show how Brazilians initially perceived an expansion of time, which steadily decreased over the course of the following weeks. Complementing the European studies, a subjective expansion of time was related to negative emotions, such as loneliness. A study in Uruguay among university students [9] assessed different measures, including the passage of time (e.g., "My days pass more slowly, time extends." and "My days pass more quickly, time flies."). They reported similar results: an association of psychological distress due to COVID-19 restrictions with a felt slower passage of time, a blurred sense of time (not knowing what time or day it was), and more boredom. In a recently published longitudinal study from Germany this association of boredom as well as deteriorated emotional state with decelerated POTJs was confirmed [10].

Taken together, different international studies suggest that: (a) the subjective experience of time was perceived as distorted during the COVID-19 pandemic, and (b) the experience of social isolation, negative emotions, boredom and distress were associated with higher ratings of deceleration and/or time dragging while more positive affect, satisfaction with the social situation, and more routine led to an acceleration of subjective time. The experience of time deceleration dominanted. These empirical findings during the COVID-19 pandemic support previous findings linking negative affect to subjectively expanded duration, and a slower

passage of time [11, 12]. It also supports findings, which showed that Positive affect combined with social satisfaction led to a subjective acceleration of time passing [13].

However, none of the studies mentioned compared the data regarding the experience of time during the pandemic to data from preceding non-pandemic periods. As many participants in studies judging the passage of time over longer life spans reported time as having passed comparably quickly [14–16], the perception of time over long intervals could be generally skewed: People might feel that long intervals of time are passing differently compared to earlier periods. This implies that the existing studies carried out during the pandemic cannot completely exclude the possibility that their participants would have reported similar levels of distorted time in none-pandemic settings, too. We therefore decided to gather data applying the exact same measurements, namely passage-of-time judgments (POTJs) covering the previous year and the previous five years, as we had done in earlier studies conducted before the pandemic [15, 17] to compare these judgments with judgments during the pandemic.

Taken together, we applied this inter-subject design to investigate whether the reported perception of time-distortion really is a specific pandemic phenomenon. Given the findings presented in previous studies, we derived three concise hypotheses: We expected POTJs provided during the pandemic to indicate a slower subjective passage of time than before the pandemic (H1); we also expected most duration judgments (DJs) for the time since the beginning of the pandemic to indicate that these were perceived as longer than usual (H2). Finally, we expected that reports of negative affect and lower social satisfaction would be associated with a subjective experience of time passing slower and the respective duration being felt as longer (H3). We also inquired about approval of and compliance with the COVID19 countermeasures imposed by the German authorities and the plausibility of these measures.

## Methods

### Participants

We collected data from 512 subjects living in Germany at the time of the survey (May, 2021) using the online participant-recruitment tool Prolific (www.prolific.co). Participants who completed the questionnaire were compensated with £0.30. Twelve subjects who did not finish the survey and/or whose answers were considered peculiar (using SoSci-Surveys Time-RSI with two standard deviations (SD) as a cut off [see 18]) were excluded. The remaining 500 subjects took an average of $M = 201$ seconds ($SD = 70.21$) to complete the questionnaire. The mean age was 29.89 years ($SD = 17.44$), 279 identified themselves as male, 219 as female, and 2 as non-binary. 42.8% of the participants were students or in vocational training, 46.4% were employed, and 6.8% were self-employed. The remaining participants were spread among other categories, such as housekeeping, unemployed, or retired (multiple options could be selected).

We compared judgments of the passage of time during the previous year of the pandemic with judgments from before the pandemic to test our main hypothesis. We included data from two previous studies [15, 17] in which the POTJs during the previous year was addressed in the exactly same manner as in the present survey. Both studies started with short measures targeting subjective well-being. Some participants directly reported their POTJs for the previous year and the previous five years, while others were asked to recall memories. There was no between-group difference regarding this manipulation, which implies that all data are suitable for the comparison with spontaneous POTJs provided during the pandemic. These combined samples contain data from 755 participants with a mean age of 29.92 years ($SD = 10.87$), 520 identifying as female, 195 as male, and 40 who did not disclose their gender. 45.9% of the

subjects were students or in vocational training, 42.4% employed, 8.3% self-employed, and the rest spread among other categories (the selection of multiple options was possible).

## Materials

**Perception of time.** Regarding the passage of time (POTJ), participants answered the question "When looking back, how did the last 12 months pass by for you?" on a 7-point Likert scale ranging from "very slow" to "very fast". A POTJ for the previous five years was also assessed to compare whether a potential effect on time experience would compromise POTJs even beyond the actual period of the pandemic. Both measures were used in the exact same manner in our prior studies [15, 17].

We asked participants for their subjective durations (DJs) for the period since the first lockdown in Germany adapting the same questions used by Ogden [4]. "Since the beginning of the first lockdown due to the coronavirus here in Germany 14 months have passed. How long does this period feel?" Participants answered this question on a 5-point Likert scale ranging from (1) "a lot shorter than 14 months", (2) "rather shorter than 14 months", (3) "approximately 14 months", (4) "rather longer than 14 months", to (5) "a lot longer than 14 months".

**Perceived changes in daily life.** Changes in life are typically associated with alterations in the perception of time (e.g., [19]), and previous research during the pandemic provided initial evidence that this also applies here [4]. We therefore requested participants to rate changes in their daily lives on seven dimensions: routine, fear for ones own as well as the health of close persons, stress-levels at home and (if applicable) at work, fatigue, and worries regarding one's livelihood compared to before the pandemic. Participants answered these questions on 5-point Likert scales ranging from "the level is. . .essentially lower" to "essentially higher". Both fear- and both stress-items were combined into one composite score for analyses, with an acceptable Cronbach's $\alpha$ = .70 for changes in fear for health and Cronbach's $\alpha$ = .57 for changes in stress levels.

**Affective states.** Since recent study results suggest effects of affective states on time estimates during the pandemic (e.g., [5, 2]), we assessed several affective judgments and experiences that seem relevant regarding the experience of time during the pandemic (9 items: bored, sad, lonely, busy, stressed, content, relaxed, angry, excited). Subjects were requested to indicate their respective average levels during the pandemic compared to before the pandemic by responding on 5-point Likert scales ranging from "a lot less" to "a lot more". We also addressed the level of satisfaction with social life during the pandemic with one item used by Ogden [4], "How satisfied are you with your social life during the pandemic?" Participants responded to this question on a 5-point Likert scale with higher scores indicating higher satisfaction. All items were then included in an exploratory and factor analysis with a KMO-index of .80 and a significant Bartlett's test of sphericity, $\chi 2(36)$ = 1,195.25, $p < 0.001$. This suggests that the sample was suitable for a factor analysis [20]. After varimax rotation, two factors were extracted with a lowest factor loading of .60 (feeling angry). Factor loadings are displayed in Table 1. The coherency of content indicated that these two factors seem to mirror two different affective patterns, the first factor subsuming a rather depressive (deactivated and negatively valenced) and lonely combination of affective states and the second factor, a rather negative agitated-aggressive combination. As these factors appeared to depict patterns and not consistent scales, the values were $z$-standardized and added up, resulting in two sum scores.

**Approval of countermeasures.** We asked participants three questions about their level of approval of the COVID-19-countermeasures introduced by the German authorities: acceptance of measures ("I consider the measures that have been implemented in our country as adequate and necessary."), compliance with measures ("Overall, I strictly adhered to the

**Table 1. Factor loadings from exploratory factor analysis.**

|  | Factor depressive-lonely | Factor agitated-aggressive |
|---|---|---|
| Bored | .76 | |
| Busy | -.72 | |
| Lonely | .66 | |
| Sad | .64 | .46 |
| Content | -.60 | -.46 |
| Stressed | | .80 |
| Relaxed | | -.70 |
| Excited | | .62 |
| Angry | | .60 |

measures."), and plausibility of measures ("I think that the measures implemented are mostly plausible and logical."). All answered on 7-point Likert scales with higher values indicating higher levels of agreement. A composite score was calculated with a very good Cronbach's $\alpha$ = 0.81.

## Procedure

The survey was conducted on SoSciSurvey [21] and started with information regarding protection of privacy (in accordance with the German DVSGO legislation) and about the study, stating that subjects would be asked to report their experience of time and its changes in the recent past. After providing informed consent by ticking a checkbox, the actual survey started with the POTJ for the preceding year. On the next page, we assessed the the DJ for the preceding 14 months, the time since the beginning of the pandemic in Germany. The POTJ for the preceding five years was inserted on the subsequent page, followed by one item asking about life satisfaction. Questions regarding the level of approval of countermeasures were asked afterwards. We finally requested demographic data, including the number of co-residents, number of children living in the same household, population of hometown (under 10,000, between 10,000 and 100,000, and more then 100,000 people), occupational status (e.g., employed, self-employed, household), and formal education.

## Preregistration and ethics

This study was conducted in accordance with the Declaration of Helsinki and the Ethics Guidelines of the University of Regensburg. These types of psychological studies of do not require the ethical approval of an Ethics Committee in Germany (see https://www.dfg.de/foerderung/faq/geistes_sozialwissenschaften/). Participants were informed about the aim of the study and gave informed consent regarding the use of the data by clicking a checkbox. This was conducted in accordance with the German DSVGO-Legislation. The study's main hypothesis was preregistered using the open science framework (https://osf.io/r29hq). All data exclusions, measured variables, and the complete procedure were reported [22]. Both data and code of the reported analyses can be accessed via an online repository on osf (https://osf.io/kc86r/).

## Statistical analyses

The statistical data analyses were conducted with SPSS® (IBM SPSS Statistics, Version 25.0). Pearson product-moment correlations were calculated for correlational analyses. Three multiple regressions were performed with the forced-entry method to analyse predictors of POTJs,

DJs, and approval of countermeasures. For POTJs and DJs, living alone as a potentially relevant demographic variable, changes in levels of fear for health, changes in stress levels due to changes in routines, changes in fatigue, and worries regarding livelihood were entered in a first step and the two emotional-reaction patterns in the second step. In the third regression with compliance as an outcome variable, the three demographic variables education level, living alone, and population of hometown were included in a first step. These were followed by changes in routine and stress and fear levels in a second step, time estimates in a third step, and the emotional-reaction pattern in a fourth step. Bootstrapping was applied in all models due to violations of assumptions [23].

## Results

Passage of time judgments (POTJs) over the preceding 12 months differed significantly between participants judging the interval before ($M$ = 5.81; $SD$ = 1.31) and those who reported their experience during ($M$ = 5.06; $SD$ = 1.56) the pandemic, $t(937.70)$ = 8.93, $p$ < .001, $d$ = 0.53. Judgments referring to the pandemic period generally indicated a slower experience of the passage of time. As Fig 1 illustrates, there was quite a substantial change in the pattern. While before the pandemic the distribution of POTJs for the preceding year was unimodal with the center of the distribution indicating a very fast passage of time, during the pandemic this transformed to a bimodal distribution with one point of highest density indicating a rather fast passage of time experienced and a second one indicating a relatively slow passage of time experienced. Nevertheless, most participants reported fast POTJs.

The proportion of responses indicating time having passed 'very fast' was 40.8% before, but only 18.8% during the pandemic. The rate of all selected options indicating a slow passage of time (values 1 to 3) rose from only 7.5% before to 20.8% during the pandemic. POTJs for the preceding five years did differ between participants from before ($M$ = 5.36, $SD$ = 1.20) and during ($M$ = 5.20, $SD$ = 1.21) the pandemic, too, but to a smaller degree ($t(1253)$ = 2.31, $p$ = .02, $d$ = 0.13). This suggests that POTJs for intervals markedly longer than the pandemic were less affected by this situational shift.

Fig 2 shows the distribution of responses for the question about how long the last 14 months had felt, i.e., since the beginning of the pandemic. Only 9.2% of participants felt that

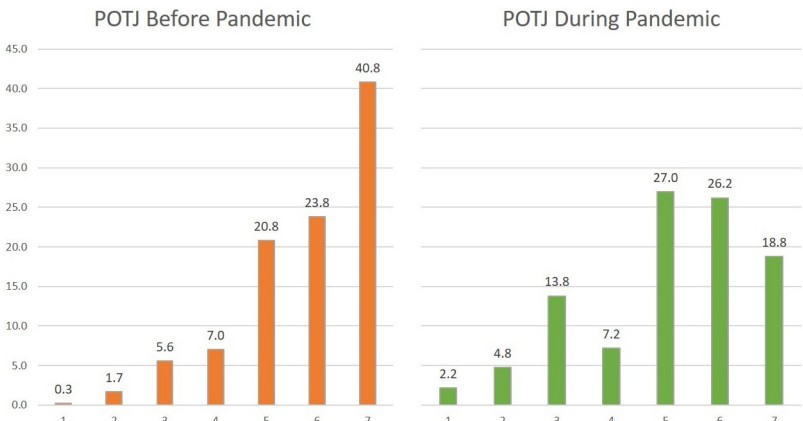

**Fig 1. Proportions of selected options (in percents) for passage of time judgments (POTJs) for the preceding year from data gathered before the pandemic (POTJs Before) and during the pandemic (POTJs During).** Values ranged from 1 = very slow to 7 = very fast judgments.

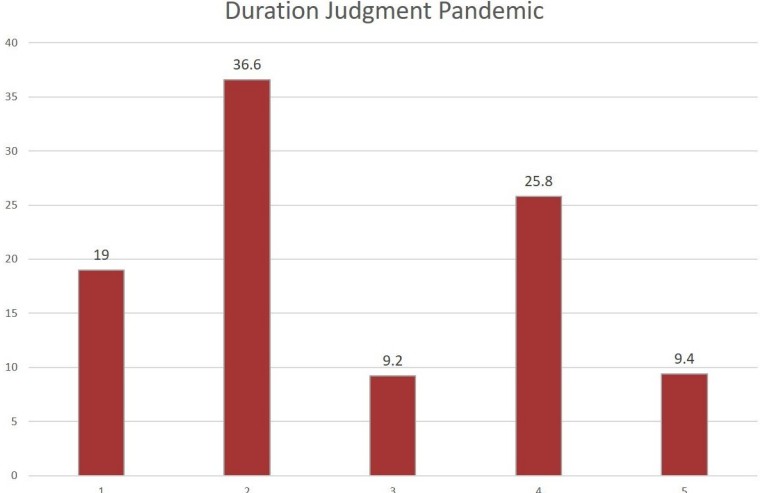

**Fig 2. Proportions of selected options (in percents) for the experienced duration over the preceding 14 months.**
Values ranged from 1 = a lot shorter to 5 = a lot longer than usual 14 months.

the 14 previous months had lasted subjectively like a typical 14-month period (value 3); 55.6%
felt that these 14 months had seemed a lot or somewhat shorter than usual (values 1, 2); 35.2%
reported that the these 14 months had seemed a lot longer or somewhat longer than usual (values 4, 5).

Table 2 shows Pearson correlation coefficients for the relationships between both subjective
duration and the POTJs for the preceding year and measures of satisfaction with social life (1
variable), emotions (9 variables), and routine/stress/worries (7 variables). A subjectively longer
DJ for the preceding 14 months was related to less satisfaction with social life, more boredom,
more sadness, less emotional satisfaction, less relaxation, higher levels of anger, and more
fatigue. Faster POTJs for the last 12 months were reported by participants who were more satisfied with their social lives.

Fear for one's own health and for the health of close friends and relatives were the only
emotional variables that were not associated with any of the two measures for the subjective
experience of time. However, these two variables were clearly associated the three measurements regarding approval with the COVID-19 countermeasures introduced in Germany,
namely acceptance (overall $M = 5.06$, $SD = 1.56$), compliance ($M = 5.72$, $SD = 1.30$), and plausibility ($M = 4.77$, $SD = 1.63$, see Table 3 for a comprehensive view). Additionally, higher levels
of anger were clearly associated with less acceptance of, less compliance with, and less perceived plausibility of these countermeasures.

Results of the two-level regression analysis for DJs are displayed in Table 4. The complete
second model, which produced the better fit, explained 7.6% of the variance, $F(8, 489) = 6.10$,
$p < .001$. The coefficient for changes in daily routines indicates that higher levels of daily routine led to shortened DJs. High values in both affective patterns predicted a perceived extension of DJs. The coefficient for fatigue was no longer significant in the second model after
inclusion of the affective patterns.

The two-level regression analysis for the POTJs was not significant, $F(6, 491) = 1.77$, $p = .103$, implying that our chosen predictors had no influence on the outcome variable, namely
the POTJs for the preceding 14 months.

The analysis of the four-level regression for approval of countermeasures revealed two significant models, the second, $F(8, 489) = 8.00$, $p < .001$ and the fourth model, $F(12, 485) =$

**Table 2. Pearson correlation coefficients for the relationships between the subjective durations of the last 14 months, POTJs for the last 12 months, and measures of emotion (9 variables) and routine/stress/worries (7 variables) across all 500 participants.**

|  | Duration last 14 months | POTJ last 12 months |
|---|---|---|
| Satisfaction with social life | -.26*** | .23*** |
| Bored | .19*** | -.15** |
| Sad | .23*** | -.12* |
| Lonely | .13** | -.08 |
| Busy | .04 | .08 |
| Stressed | .13** | .04 |
| Content | -.20*** | .18*** |
| Relaxed | -.19*** | .14** |
| Angry | .23*** | -.12* |
| Excited | .05 | -.02 |
| Routine in daily life | -.12* | .12* |
| Fear for my health | .09 | -.04 |
| Fear for health of people close to me | .08 | .00 |
| Stress at home | .09 | -.04 |
| Stress at work | .09 | .02 |
| Fatigue | .19*** | -.07 |
| Worry about my/our livelihood | .12* | -.07 |

*** $p < .001$,

** $p < .01$,

* $p < .05$,

all $p$-levels (two-tailed) have been adjusted to correct for alpha-error inflation using the Benjamini-Hochberg False-Discovery Rate [24] with an online tool by Hemmerich [25].

6.25, $p < .001$. The fourth model explained the largest amount of variance, corrected $R^2 = .11$, as can be seen in Table 5, which depicts both significant models. While the time estimates had no impact on the outcome variable, 'fear of infection' predicted higher compliance, and 'worries regarding livelihood' predicted lower compliance. The coefficient for 'changes in daily stress levels' was no longer significant when the affective patterns were included in the model. Lower levels of the agitated-aggressive pattern predicted higher levels of approval of the countermeasures. In the fourth model, the coefficient for fatigue ($p = .051$) also came close to the threshold of significance.

## Discussion

This study, which investigated time perception during the pandemic in Germany, led to several results: (1) In May 2021, people reported the previous year as having passed markedly slower than people asked before the pandemic. (2) The experienced duration of the previous 14 months following the start of the pandemic showed a bimodal distribution with a majority of participants rating these to have seemed relatively shorter. A minority of participants rated these as relatively longer, i.e., people were somewhat split in their judgments of experienced duration. (3) The experience of time having passed fast during the previous year was mirrored by a rating indicating a shortened perception of the 14-month time interval. (4) Both negative affective patterns, 'depressive-lonely' and 'agitated-aggressive' predicted the DJs in the way that high levels of negativity and increased fatigue led to a felt extension of time, regardless of

**Table 3. Pearson correlation coefficients for the acceptance of, compliance with, and subjective plausibility of COVID-19 countermeasures with measures of emotion (9 variables) and routine/stress/worries (7 variables) across all 500 participants.**

|  | Acceptance | Compliance | Plausibility |
|---|---|---|---|
|  | Covid Measures | Covid Measures | Covid Measures |
| Satisfaction with social life | .12* | -.01 | .12* |
| Bored | -.03 | -.02 | -.01 |
| Sad | -.07 | .03 | -.05 |
| Lonely | -.05 | .07 | -.08 |
| Busy | .02 | -.01 | -.01 |
| Stressed | -.11 | .02 | -.11* |
| Conten | .10 | -.08 | .08 |
| Relaxed | .08 | -.02 | .07 |
| Angry | -.17** | -.01 | -.23*** |
| Excited | -.04 | .04 | -.08 |
| Routine in daily life | .06 | -.02 | .09 |
| Fear for my health | .16** | .22*** | .19*** |
| Fear for health of people close to me | .27*** | .22*** | .23*** |
| Stress at home | -.10 | .03 | -.11* |
| Stress at work | -.08 | .00 | -.05 |
| Fatigue | -.02 | .08 | .01 |
| Worry about my/our livlihood | -.08 | .00 | -.09 |

*** $p < .001$,

** $p < .01$,

* $p < .05$,

all p-levels (two-tailed) have been adjusted to correct for alpha-error-inflation using the Benjamini-Hochberg-False-Discovery-Rate [24] with an online tool by Hemmerich [25].

activation levels. A perceived shortening of Duration was associated with a perceived increase in routine. (5) Regarding the governmental countermeasures, the more people feared for their health and the health of people close to them, the more they approved of the countermeasures. The 'agitated-aggressive' affective pattern and 'worries regarding livelihood' were associated with lower levels of approval of countermeasures.

Our main result is that changes in the perceived passage of time during the pandemic, as concluded by many authors in several studies from different countries (e.g., [2, 3, 4, 7, 9]), can be confirmed even when comparing the passage of the previous year to a baseline: the previous year was generally judged as having passed markedly slower by participants who provided their judgments during the pandemic (note that the pandemic covered this whole year) compared to participants who made these judgments before the pandemic. This corresponds with results from a UK study, where almost 40% of the participants reported the time since the beginning of the pandemic (8 months) as longer than a normal 8-month period [4]. The duration of the previous 14 months (the time since the pandemic reached Germany and a first lockdown was installed) revealed a split pattern with relatively more participants reporting time having shortened (55.6%) during the pandemic, while fewer people reported these 14 months as having felt longer (35.2%) than a normal 14-month period. This is almost exactly opposite to the pattern found in the UK [4], where 54% of the participants rated the time since the beginning of the pandemic (8 months) as subjectively longer and only 29% as shorter than a typical 8-month period. This difference in the response patterns could have been caused by different public responses to the pandemic, but it might also reflect a methodological artefact: In

**Table 4. Linear model of predictors of DJs for the last 12 months.**

| Model | Predictors | Model for POTJ | | |
|---|---|---|---|---|
| | | *b* | *SE B* | *β* |
| 1 | Constant | 1.52 (0.64, 2.40) | .45 | |
| | Living alone | 0.05 (-0.22, 0.32) | .13 | .02 |
| | Change in routine | 0.11 (0.02, 0.19) | .05 | .10* |
| | Fear for health | -0.14 (-0.31, 0.04) | .09 | -.07 |
| | Changes in daily stress | -0.04 (-0.20, 0.11) | .08 | -.03 |
| | Fatigue | -0.20 (-0.33, -0.06) | .07 | -.14* |
| | Worries regarding livelihood | -0.05 (-0.18, 0.10) | .07 | -.03 |
| | Corrected $R^2$ | .04*** | | |
| 2 | Constant | 2.33 (1.38, 3.29) | .49 | |
| | Living alone | 0.05 (-0.19, 0.30) | .12 | .02 |
| | Change in routine | 0.11 (0.01, 0.20) | .05 | .10* |
| | Fear for health | -0.10 (0.27, 0.08) | .09 | -.05 |
| | Changes in daily stress | 0.04 (-0.13, 0.20) | .09 | .02 |
| | Fatigue | -0.14 (-0.27, 0.00) | .07 | -.10 |
| | Worries regarding livelihood | 0.01 (-0.13, 0.14) | .07 | .00 |
| | Affective Pattern: depressive-lonely | -0.05 (-0.09, -0.02) | .02 | -.14* |
| | Affective Pattern: agitated-aggressive | -0.06 (-0.10, 0.01) | .03 | -.12* |
| | Corrected $R^2$ | .08*** | | |
| | $\Delta R^2$ | .04 | | |

*Note.* Coefficients were bootstrapped with bias correlated and accelerated confidence intervals.

Significant *β-values* are indicated

*$p < .05$,

***$p < .001$.

the UK-study, participants were first asked to rate the duration of the previous day and the previous week. In both questions, participants were explicitly requested to rate their experience of time "passing in comparison with normal (i.e., before lockdown)?" [4, p. 6]. Although this comparison was not addressed in the question regarding the preceding eight months, it seems likely that participants compared these eight months with the same interval before the pandemic. In our study, an explicit comparison with the same interval before the pandemic was never requested. Thus, many more people judged the interval as relatively short—in line with the POTJs—but a relatively large number of responses indicated a prolonged time experience. In fact, the substantial association between POJTs and DJs for the intervals covered by the pandemic suggests that both measures are relatively equal concerning retrospective judgments for these long intervals. This adds support to the finding that POTJs and DJs for longer intervals starting in the range of several minutes coincide [26], although this relation can not be found under all circumstances [27].

By comparing POTJs for the last year during the pandemic with POTJs from earlier, we found compelling support for pre-existing evidence regarding the experience of a lenghtened/ slowed experience of time for the interval following the beginning of the pandemic as compared to earlier times, e.g., [3, 4, 8, 9, 20]. Therefore, we can exclude the possibility that previous results, which reported distorted time-experiences during the pandemic, reflect only the fact that the experience of time for long intervals feels generally skewed for most people (i.e., under non-pandemic circumstances only few people report their experience of time as neither fast nor slow, too (e.g., [14–16]).

**Table 5. Linear model of predictors of approval of countermeasures.**

| Model | Predictors | Model for approval of countermeasures | | |
|---|---|---|---|---|
| | | *b* | *SE B* | *β* |
| 2 | Constant | 3.13 (2.02, 4.24) | .56 | |
| | Population of hometown | -0.07 (-0.22, 0.08) | .08 | -.04 |
| | Living alone | 0.06 (-0.04, 0.15) | .05 | .05 |
| | Educational level | 0.11 (-0.03, 0.26) | .07 | .07 |
| | Change in routine | 0.03 (-0.06, 0.11) | .05 | .03 |
| | Fear for health | 0.60 (0.40, 0.78) | .09 | .31* |
| | Changes in daily stress | -0.15 (-0.29, -0.01) | .07 | -.09* |
| | Fatigue | 0.09 (-0.04, 0.22) | .06 | .06 |
| | Worries regarding livelihood | -0.18 (-0.31, -0.05) | .07 | -.13** |
| | Corrected $R^2$ | .10*** | | |
| 4 | Constant | 2.47 (0.98, 3.83) | .72 | |
| | Population of hometown | -0.82 (-0.23, 0.07) | .08 | -.05 |
| | Living alone | 0.06 (-0.04, 0.14) | .05 | .05 |
| | Educational level | 0.13 (-0.02, 0.30) | .08 | .08 |
| | Change in routine | 0.03 (-0.06, 0.12) | .05 | .03 |
| | Fear for health | 0.62 (0.43, 0.81) | .09 | .32*** |
| | Changes in daily stress | -0.05 (-0.21, 0.12) | .08 | -.03 |
| | Fatigue | 0.12 (0.01, 0.25) | .06 | .09 |
| | Worries regarding livelihood | -0.16 (-0.30, -0.03) | .07 | -.12* |
| | POTJ | 0.01 (-0.08, -0.09) | .04 | .01 |
| | DJ | -0.01 (-0.11, 0.10) | .05 | -.01 |
| | Affective Pattern: depressive-lonely | -0.00 (-0.04, 0.03) | .02 | -.01 |
| | Affective Pattern: agitated-aggressive | -0.07 (-0.12, -0.02) | .03 | -.16** |
| | Corrected $R^2$ | .11** | | |
| | $\Delta R^2$ | .02 | | |

*Note.* Coefficients were bootstrapped with bias-correlated and accelerated confidence intervals.

Significant *β-values* are indicated

* = $p < .05$,

** = $p < .01$,

*** = $p < .001$.

Regarding potential explanations for this relative deceleration, we found that higher levels of emotional negativity predict an experience of time as having passed slower for the preceding 12 months in our German sample. When investigating the effects of specific emotions, high boredom and low levels of social satisfaction, contentment and relaxation had the highest associations with a perceived deceleration of time. High boredom, anger and sadness, as well as low levels of contentment, satisfaction and relaxation were associated with a longer DJ. These results, in particular the association of asubjective deceleration as well as longer duration with negative affect are in line with findings from studies conducted in Italy [2], France [5–7], the UK [3, 4], Brazil [8], Uruguay [9], and Germany [10]. This finding is, therefore, very robust and largely independent of the cultural backgrounds.

How can these consistent findings be explained? The most common approach to explain differences in retrospective time experience is to link it to experiences of contextual changes in the regarding intervals and the resulting storage size, which reflects the quantity and complexity of stored information from the respective interval [19, 28]. According to such memory-

based approaches, intervals filled with multiple and diverse experiences lead to more memory content, which results in the impression of time being rich in memories and perceived as long. These theories suggest that the experience of time is, other than actual sensory experiences, approached through the representation of images from the past [29], an idea that was proposed already in the 19th century by William James (see [30], for a more detailed discussion). Routine, which implies a life with few changes and therefore potentially few differing memories, has been shown to be associated with a faster passage of time for longer time intervals [31] (especially studies 5 and 6). This may offer a plausible explanation for the pandemic situation, where, due to lockdowns and social-distancing measures, the diversity in possible activities and life events was heavily restricted, potentially leading to fewer accessible memories. Following these deliberations, the time since the beginning of the pandemic should have been perceived as having shortened/passed faster than usual for most people. This was indeed often reported in the media relatively early in the pandemic, see [1]. However, our data, as well as the preceding studies, revealed a different picture with a notable tendency towards a relative deceleration of time passage compared to before the pandemic. At the same time and similar to results provided by Ogden [4], we found an association between routine in daily life and the experienced duration, but only for the measure targeting experienced duration, not for POTJs, and this correlation is very small. This seems plausible given recent findings that found no support for an association between life events and/or important memories with POTJs for intervals spanning over several years [32].

Patterns of negative affect have proven to be better predictors for the experience of time in both our and preceding studies. Before the pandemic, only few studies were conducted that investigated the interplay between affect and the experienced time over longer intervals, e.g., in the range of many minutes. In these studies, depression, which is by definition associated with low levels of well-being [33], is clearly linked to a perceived deceleration of time (for an overview see [34]). Additionally, some studies that investigated shorter intervals in the range of seconds provided evidence showing that intervals filled with unpleasant stimuli are perceived as longer or that people in negative affective states have an extended experience of time (for an overview see [35]). Another interesting association comes from previous studies investigating the interplay between emotion and waiting. These show that negative emotions experienced while waiting lead to an overestimation of the waiting period [36, 37] while positive affect is linked to the experience of accelerated time (e.g., higher levels of relaxation or social satisfaction are associated with a perception of time as having been shorter/passed more quickly). These results suggest that affect matters for retrospective (i.e., when people judge their experience after the interval in question) judgments of the experience of time.

In contrast, for prospective time experience (i.e., judgments of time during the experienced interval), the association between attention allocation and time experience has been repeatedly discussed and theories postulated that aligning attention towards time leads to an extension of the subjective experience of time [38]. The consistent results of positive affect being associated with experienced faster time passage/shorter time experience and vice versa for negative affect might suggest that a similar mechanism is also at work here. People probably focus more often on time and wish for it to pass during an interval associated with high levels of negative affect and aversive states. In such situations, time drags and feels as if it were passing too slowly. During relatively more pleasant situations, there is no craving for time to pass faster since one is occupied with absorbing activities. Thus, the split result regarding the duration of the time since the beginning of the pandemic could reflect that some people experienced time as dragging, while others were able to adapt better and experienced the situation as less aversive or even partially positive, which results in time experienced as shorter than usual (note that most

adults consistently rate long intervals ranging from a day up to several years as having passed quickly under usual circumstances, e.g., [14, 39, 16].

These deliberations are still tentative because, to the best of our knowledge, concise theories explaining an association between low well-being / negative affect and a deceleration in subjective time for retrospective judgments over long intervals in the range of months and years are missing to date. A more elaborated explanation of these findings is a matter for future research. We can presently only conclude that our study, in line with previous ones, highlights that the pandemic has led to a relative deceleration of the subjective experience of time and that the experience of time during the pandemic was markedly associated with social satisfaction and the affect people have reported for the time since the beginning of the pandemic.

Additionally, it is worth noting that our regression-models for the DJs explained only a small proportion of variance despite a relatively large sample size. This implicates that there are many other factors, which potentially impact ratings of time-experience. Although we tried to capture some variables, which address life-circumstances and the respective changes during the pandemic, our study did not target these in depth. It seems, for example, likely that higher demands in childcare and the resulting increases in workload might have led to variations in the experience of time and disparities between men and women, since numerous studies revealed that the decrease in childcare at home due to the discontinuation of childcare facilities and schools during the pandemic was largely shouldered by women (e.g., [40–42]). This seems relevant for the experience of time since previous research also revealed that parents report a faster passage of time compared to adults without children [43]. One might also argue that items such as worries regarding ones livelihood depict underlying phenomena only on a shallow level, since such worries could be relatively independent of an existential dimension, which in turn might be associated with factors having a potential impact on the experience of time (e.g., depression). However, these items were included in our study in order to detect potential domains that might have relevance for the experience of time. Potential associations uncovered here might serve as starting points for future research, which targets the interplay of demographical and situational factors with the experience of time in depth.

Among our regression-models, an exploratory analysis for predicting approval of the COVID-19 countermeasures introduced by the German authorities led to the highest proportion of explained variance. With this analysis, we confirmed previous findings that high levels of fear for health were associated with higher reports of compliance, plausibility, and agreement with the restrictions in other countries [44, 45]. In contrast, participants who felt their livelihoods threatened reported lower levels of approval with the measures. This emphasizes the importance of specific fears and worries for approval of restrictions and countermeasures during a pandemic. Governmental policies should point out the danger posed by a virus during a pandemic, but also address worries of the security of livelihoods because both influence approval of the countermeasures. Our second finding showed that approval of countermeasures was also associated with higher levels of agitated-aggressive affects. This complements previous theories predicting compliance with countermeasures which focus on the effects of rather stable variables, such as demographics (e.g., older age, female gender; [46]), personality (e.g., low neuroticism and high conscientiousness; [45]), or trust in science [47] and governments [48]. Our study results amend these findings with an affective-state component.

Summed up, our study provides a number of important findings. Corresponding with previous research, we found (a) a relative deceleration of experienced time during the pandemic. Notably, we demonstrated this by directly comparing POTJ-ratings from during the pandemic to ratings from before the pandemic. Our data also supports previous findings showing that (b) an extended time perception was linked to negative affect. A subset of negative affect, namely negative-aggressive emotions, was (c) clearly linked to less approval of the

governmental countermeasures introduced to overcome the pandemic, while (d) health-related fear predicts higher levels of approval of these measures.

## Supporting information

**S1 File.**
(ZIP)

## Author Contributions

**Conceptualization:** Ferdinand Kosak, Marc Wittmann.

**Data curation:** Ferdinand Kosak, Iris Schelhorn.

**Formal analysis:** Ferdinand Kosak, Iris Schelhorn, Marc Wittmann.

**Methodology:** Ferdinand Kosak, Marc Wittmann.

**Project administration:** Marc Wittmann.

**Software:** Ferdinand Kosak.

**Supervision:** Marc Wittmann.

**Writing – original draft:** Ferdinand Kosak, Iris Schelhorn, Marc Wittmann.

**Writing – review & editing:** Ferdinand Kosak, Iris Schelhorn, Marc Wittmann.

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
