## [Decision Letter · Decision Letter 0]

16 Mar 2022

PONE-D-21-39406The Subjective Experience of Time during the Pandemic in Germany: the big SlowdownPLOS ONE

Dear Dr. Kosak,

Thank you for submitting your manuscript to PLOS ONE. After careful consideration, we feel that it has merit but does not fully meet PLOS ONE’s publication criteria as it currently stands. Therefore, we invite you to submit a revised version of the manuscript that addresses the points raised during the review process. Both Reviewers agree that your manuscript can be published but they have a few suggestions to further improve it. All issues raised are relatively minor, but I encourage you to carefully consider them.

We look forward to receiving your revised manuscript.

Kind regards,

Enrico Toffalini, Ph.D

Academic Editor

PLOS ONE

Journal Requirements:

Reviewers' comments:

Reviewer's Responses to Questions

**Comments to the Author**

1. Is the manuscript technically sound, and do the data support the conclusions?

Reviewer #1: Yes

Reviewer #2: Yes

2. Has the statistical analysis been performed appropriately and rigorously? 

Reviewer #1: Yes

Reviewer #2: Yes

3. Have the authors made all data underlying the findings in their manuscript fully available?

Reviewer #1: Yes

Reviewer #2: Yes

4. Is the manuscript presented in an intelligible fashion and written in standard English?

Reviewer #1: Yes

Reviewer #2: Yes

5. Review Comments to the Author

Reviewer #1: The current study investigated subjective time during the Covid 19 pandemic. The passage of time judgments could be compared to similar judgments gathered before the pandemic; authors found that the previous year passed slower during the pandemic. On the other hand, the experienced duration of 14 months revealed mixed results in the form of bimodality around the target. Negative emotions and social isolation were associated with the slower passage of time. The exploratory analysis revealed some of the affective state predictors of compliance and approval of government imposed countermeasures.

This is a straightforward study and a nice effort to address how subjective time is affected by the pandemic and the factors that mediate this potentially altered sense of time. I have a few comments only.

Models account for only a small proportion of variance. I think authors should clearly discuss this issue in light of using a large sample size.

Lines 257-260: Figure 1. What is off/unexpected here is the pre-pandemic data. This is, if anything, normalized during the pandemic. The authors should discuss this.

Regarding the bimodality judgment of time; is it possible that participants were simply meeting the experimenters’ expectations given the way the question was asked? This could explain the escape from the middle. In this case, what would be most informative is the difference between the data for the corresponding absolute deviations from the middle.

Lines 126-135 are repetitive given the earlier text

Typos:

Line 120: delete "in"

Line 160: use space

Line 406: therefore

Reviewer #2: In the manuscript, "The Subjective Experience of Time during the Pandemic in Germany: the big Slowdown", the authors address an interesting phenomenon regarding the experience of time during the pandemic. The manuscript clearly addresses issues that go beyond the pandemic, regarding psychological well-being and one's perception of time and the role that being active/busy plays in how one passes time that could have far-reaching implications in how we address symptoms of depression/agitation, etc.

Overall, I think the manuscript is well-written and significantly contributes to the literature. I have some very minor edits followed by one more substantial edit.

Minor Edits

Page 10, Line 105: A space needs to be added between routine and led

Page 11, Line 144: diviations should be spelled "deviations" for this journal

Page 12, Line 160: A space should be added between time and perception

Pages 17 & 18, Table: livlihood should be spelled livelihood

Major Concern

I understand that the authors included concerns regarding livelihood, but I feel like this does not adequately address the role of socioeconomic status. Is it not true that one could be earning a substantial income and have significant savings and be equally concerned about their job (while clearly able to continue to pay bills during the pandemic) versus someone who is living paycheck to paycheck and has the same concerns about their livelihood, but with significantly more devastating implications, potentially increasing depression, agitation, etc.?

I do not expect the authors to try to gather this information at this point, but it would be helpful to address it in more depth. In the same vein, it would be helpful to address the limitation that the manuscript does not address issues related to gender/childcare. For example, being busy with work or social activities might result in different perceptions of time than being busy with childcare (that was not chosen or resulted in loss of work-related satisfaction).

Neither of these issues make the manuscript unpublishable, but mentioning these issues in terms of limitations would be appreciated.

6. PLOS authors have the option to publish the peer review history of their article (what does this mean?). If published, this will include your full peer review and any attached files.

Reviewer #1: No

Reviewer #2: No

---

## [Author Response · Author response to Decision Letter 0]

6 Apr 2022

Response to Reviews:

Dear Reviewers,

thank you for the thorough reading and helpful comments on our manuscript. These helped us to improve our work. You find our responses to your comments below.

Please note that we found a mistake regarding our regression models: the models regarding duration judgments and passage of time judgments were interchanged in our original version of the manuscript. This as well as the according references in the subsequent discussion have been corrected. 

Additionally, we included one new study in our introduction (lines 99-101), which has been published after the initial submission of our manuscript. 

With regards,

Ferdinand Kosak, Iris Schelhorn, & Marc Wittmann

Review 1:

1. Models account for only a small proportion of variance. I think authors should clearly discuss this issue in light of using a large sample size.

We thank the reviewer for highlighting this important issue. We added a paragraph in the discussion, addressing this topic. 

It reads as follows (lines 468ff):

“Additionally, it is worth noting that our regression-models for the DJs explained only a small proportion of variance despite a relatively large sample size. This implicates that there are many other factors, which potentially impact ratings of time-experience.”

In response to Reviewer 2, we continue this paragraph by discussing some of these potential factors.

2. Lines 257-260: Figure 1. What is off/unexpected here is the pre-pandemic data. This is, if 

anything, normalized during the pandemic. The authors should discuss this.

We thank the reviewer for pointing out that the pre-pandemic data were somewhat unexpected. We want to highlight that this comparison is the fundamental idea of our study, since it enables us to compare judgments during the pandemic with a baseline, which is not present in all of the previous studies.

As suggested by the reviewer (and as addressed in response to the reviewers comment #4), the experience of time is likely to be generally skewed, independently from pandemic or non-pandemic times (we discuss this in line 116ff). We want to argue that it is unlikely that there is something like a “normal” experience of time for long intervals (i.e., much longer than several seconds). The subjective reports are likely to be a result of cognitive mechanisms and other influences, e.g., affective components, as highlighted in this manuscript. Therefore, it is not our aim to label either situational circumstances (pandemic or non-pandemic) as the ones where the time-experience is normal. We rather utilize the pandemic circumstances in order to increase the understanding of the time-experience for long intervals.

We tried to highlight the relevance of our approach again (in addition to line 114ff.) by amending the discussion with the following sentence (lines 384-388):

“Therefore, we can exclude the possibility that previous results, which reported distorted time-experiences during the pandemic, reflect only the fact that the experience of time for long intervals feels generally skewed for most people (i.e., also under non-pandemic circumstances only few people report their experience of time as neither fast nor slow; see, e.g., Friedmann & Janssen, 2010; Kosak et al., 2019; Wittmann & Lehnhoff, 2005).”

3. Regarding the bimodality judgment of time; is it possible that participants were simply meeting the experimenters’ expectations given the way the question was asked? This could explain the escape from the middle. In this case, what would be most informative is the difference between the data for the corresponding absolute deviations from the middle.

We thank the reviewer for raising this important question. However, our two measures regarding the experience of time (POTJ and DJ) and the respective items were carefully selected to avoid such an experimenter-expectation-effect with available center values indicating the POTJ as “neither fast nor slow” and the DJ as “approximately 14 months” respectively (see lines 163-175). 

In contrast to the two studies from the UK, we did not explicitly request our participants to contrast the current time-experience with the experience from before the pandemic, which might suggest to participants that the experimenter expects changes compared to the ‘normal’ experience.

Additionally, we want to highlight that the POTJs have been used in our previous studies using the exact same wording. These are the standard questions employed in studies on the experience of time. In the data from these studies, no bimodal pattern was present.

4. Lines 126-135 are repetitive given the earlier text

We thank the reviewer for pointing out potential redundancies. We rephrased the paragraph and cut it slightly back in order to highlight and summarize the concrete hypotheses, which are to be investigated in this study. It now reads as follows (now lines 128-138):

“Taken together, we apply this inter-subject design to investigate whether the reported perception of time-distortion really is a specific pandemic phenomenon. Given the findings presented in previous studies, we derived three concise hypotheses: We expected POTJs provided during the pandemic to indicate a slower subjective passage of time than before the pandemic (H1); we also expected most duration judgments (DJs) for the time since the beginning of the pandemic to indicate that these were perceived as longer than usual (H2). Finally, we expected that reports of negative affect and lower social satisfaction would be associated with a subjective experience of time passing slower and the respective duration being felt as longer (H3). We also inquired about approval of and compliance with the COVID19 countermeasures imposed by the German authorities and the plausibility of these measures.”

5. Typos

We thank the reviewer for the careful reading of the manuscript. The typos have been corrected.

Review 2:

Minor Edits:

We thank the reviewer for the thorough reading, we corrected all mistakes found by the reviewer.

Major Issue:

I understand that the authors included concerns regarding livelihood, but I feel like this does not adequately address the role of socioeconomic status. Is it not true that one could be earning a substantial income and have significant savings and be equally concerned about their job (while clearly able to continue to pay bills during the pandemic) versus someone who is living paycheck to paycheck and has the same concerns about their livelihood, but with significantly more devastating implications, potentially increasing depression, agitation, etc.? 

I do not expect the authors to try to gather this information at this point, but it would be helpful to address it in more depth. In the same vein, it would be helpful to address the limitation that the manuscript does not address issues related to gender/childcare. For example, being busy with work or social activities might result in different perceptions of time than being busy with childcare (that was not chosen or resulted in loss of work-related satisfaction).

We thank the reviewer for pointing us to these limitations. We addressed these by adding a new paragraph (lines 470-473), which reads as follows:

“Although we tried to capture some variables, which address life-circumstances and the respective changes during the pandemic, our study did not target these in depth. It seems, for example, likely that higher demands in childcare and the resulting increases in workload might have led to variations in the experience of time and disparities between men and women (footnote 6). In fact, previous research revealed that parents experience a faster passage of time in normal circumstances than comparable adults without children (Wittmann & Mella, 2021). One might also argue that items such as worries regarding ones livelihood depict underlying phenomena only on a shallow level, since such worries could be relatively independent of an existential dimension, which in turn might be associated with factors having a potential impact on the experience of time (e.g., depression). However, these items were included in our study in order to detect potential domains that might have relevance for the experience of time. Potential associations uncovered here might serve as starting points for future research, which targets the interplay of demographical and situational factors with the experience of time in depth.”

footnote 6: Numerous studies revealed that the decrease in childcare at home due to the discontinuation of childcare facilities and schools during the pandemic was largely shouldered by women (e.g., de Gennaro et al., 2022; Pabilonia & Vernon, 2022; Petts et al., 2020),

---

## [Decision Letter · Decision Letter 1]

14 Apr 2022

The Subjective Experience of Time during the Pandemic in Germany: the big Slowdown

PONE-D-21-39406R1

Dear Dr. Kosak,

We’re pleased to inform you that your manuscript has been judged scientifically suitable for publication and will be formally accepted for publication once it meets all outstanding technical requirements.

Kind regards,

Enrico Toffalini, Ph.D

Academic Editor

PLOS ONE

Additional Editor Comments (optional):

Reviewers' comments:

Reviewer's Responses to Questions

**Comments to the Author**

1. If the authors have adequately addressed your comments raised in a previous round of review and you feel that this manuscript is now acceptable for publication, you may indicate that here to bypass the “Comments to the Author” section, enter your conflict of interest statement in the “Confidential to Editor” section, and submit your "Accept" recommendation.

Reviewer #1: All comments have been addressed

Reviewer #2: All comments have been addressed

2. Is the manuscript technically sound, and do the data support the conclusions?

Reviewer #1: Yes

Reviewer #2: Yes

3. Has the statistical analysis been performed appropriately and rigorously? 

Reviewer #1: Yes

Reviewer #2: Yes

4. Have the authors made all data underlying the findings in their manuscript fully available?

Reviewer #1: Yes

Reviewer #2: Yes

5. Is the manuscript presented in an intelligible fashion and written in standard English?

Reviewer #1: Yes

Reviewer #2: Yes

6. Review Comments to the Author

Reviewer #1: Authors have addressed the issues I have raised during the initial round of reviews. I think the paper is publishable in its current form.

Reviewer #2: Thank you for addressing the concerns. This is a very interesting manuscript that adds to the literature in a meaningful way.

7. PLOS authors have the option to publish the peer review history of their article (what does this mean?). If published, this will include your full peer review and any attached files.

Reviewer #1: No

Reviewer #2: **Yes: **Deana Davalos

---

## [Editor Report · Acceptance letter]

27 Apr 2022

PONE-D-21-39406R1 

The Subjective Experience of Time during the Pandemic in Germany: the big Slowdown 

Dear Dr. Kosak:

I'm pleased to inform you that your manuscript has been deemed suitable for publication in PLOS ONE. Congratulations! Your manuscript is now with our production department. 

Kind regards, 

on behalf of

Dr. Enrico Toffalini 

Academic Editor

PLOS ONE